# Test for non-negligible adverse shifts

**Vathy M. Kamulete** (ORCID)

Enterprise Model Risk Management
Royal Bank of Canada
Toronto, Canada
`vathy.kamulete@rbccm.com`

## Abstract

Statistical tests for dataset shift are susceptible to false alarms: they are sensitive to minor differences when there is in fact adequate sample coverage and predictive performance. We propose instead a framework to detect adverse shifts based on outlier scores, `D-SOS` for short. `D-SOS` holds that the new (test) sample is not substantively worse than the reference (training) sample, and not that the two are equal. The key idea is to reduce observations to outlier scores and compare contamination rates at varying weighted thresholds. Users can define what *worse* means in terms of relevant notions of outlyingness, including proxies for predictive performance. Compared to tests of equal distribution, our approach is uniquely tailored to serve as a robust metric for model monitoring and data validation. We show how versatile and practical `D-SOS` is on a wide range of real and simulated data.

## 1 INTRODUCTION

Suppose we fit a predictive model on a training set and predict on a test set. Dataset shift, also known as data or population drift, occurs when training and test distributions are not alike [Kelly et al., 1999, Quionero-Candela et al., 2009]. This is essentially a sample mismatch problem. Some regions of the data space are either too sparse or absent during training and gain importance at test time. We want methods that alert users to the presence of unexpected inputs in the test set [Rabanser et al., 2019]. To do so, a measure of divergence between training and test set is required. Can we not use the many modern off-the-shelf multivariate tests of equal distributions for this?

One reason for moving beyond tests of equal distributions is that they are often too strict. They require high fidelity between training and test set everywhere in the input domain. However, not *all* changes in distribution are a cause for concern – some changes are benign. Practitioners distrust these tests because of false alarms. Polyzotis et al. [2019] comment:

> statistical tests for detecting changes in the data distribution [. . . ] are too sensitive and also uninformative for the typical scale of data in machine learning pipelines, which led us to seek alternative methods to quantify changes between data distributions.

Even when the difference is small or negligible, tests of equal distributions reject the null hypothesis of no difference. Monitoring model performance and data quality is a critical part of deploying safe and mature models in production [Paleyes et al., 2020, Zhang et al., 2020]. An alarm should only be raised if a shift warrants intervention. Retraining models when distribution changes are benign is both costly and ineffective [Vovk et al., 2021]. To tackle these challenges, we propose `D-SOS` instead.

In comparing the test set to the training set, `D-SOS` pays more attention to the regions — typically, the outlying regions — where we are most vulnerable. To confront false alarms, it uses a robust test statistic, the weighted area under the receiver operating characteristic curve (WAUC). The weights in the WAUC discount the safe regions of the distribution. As best we know, this is the first time that the WAUC is being used as a test statistic in this context. The goal of `D-SOS` is to detect *non-negligible adverse shifts*. This is reminiscent of noninferiority tests [Wellek, 2010], widely used in healthcare to determine if a new treatment is in fact not inferior to an older one. Colloquially, `D-SOS` holds that the new sample is not substantively worse than the old sample, and not that the two are equal.

`D-SOS` moves beyond tests of equal distributions and lets users specify which notions of outlyingness to probe. The choice of the score function plays a central role in formaliz-

*Accepted for the 38th Conference on Uncertainty in Artificial Intelligence* (UAI 2022).

ing what we mean by *worse*. These scores can come from out-of-distribution (outlier) detection, two-sample classification, uncertainty quantification, residual diagnostics, density estimation, dimension reduction, and more. While some of these scores are underused and underappreciated in two-sample statistical tests, they can be more telling than the density-based scores.

In this paper, we make the following contribution. We derive `D-SOS`, a novel and robust two-sample test for no adverse shift, from tests of goodness-of-fit. The main takeaway is that given a generic method to assign an outlier score to a data point, `D-SOS` turns these scores into a two-sample test for no adverse shift. It converts arbitrary outlier scores into interpretable probabilities, namely $p-$values. The field is replete with tests of equal distribution and goodness-of-fit. We have comparatively fewer options for tests of no adverse shift. We have created an accompanying R package `dsos` for our method. In addition, all code and data used in this paper are publicly available.

## 2  MOTIVATIONAL EXAMPLE

For illustration, we apply `D-SOS` to the canonical `iris` dataset [Anderson, 1935]. The task is to classify the species of Iris flowers based on $d = 4$ covariates (features) and $n = 50$ observations for each species. We show how `D-SOS` helps diagnose false alarms. We highlight that (1) changes in distribution do not necessarily hurt predictive performance, and (2) points in the densest regions of the distribution can be the most difficult – unsafe – to predict.

We consider four tests of no adverse shift. Each test uses a different score. For two-sample classification, this score is the probability of belonging to the test set. For density-based out-of-distribution (OOD) detection, the score comes from isolation forest; this score is inversely related to the local density. For residual diagnostics, it is the out-of-sample (out-of-bag) prediction error from random forests. Finally, for confidence-based OOD detection, it is the standard error of the mean prediction from random forests, a proxy for prediction (resampling) uncertainty similar to [Schulam and Saria, 2019]. Only the first notion of outlyingness – two-sample classification – pertains to modern tests of equal distributions; the others capture other meaningful notions of adverse shifts. For all these scores, higher is *worse*: higher scores indicate that the observation is diverging from the desired outcome or that it does not conform to the training set.

For the subsequent tests, we split `iris` into 2/3 training and 1/3 test set. The train-test pairs correspond to two partitioning strategies: (1) random sampling and (2) in-distribution (most dense) examples in the test set. How do these sample splits fare with respect to the aforementioned tests? Let $s$ and $p$ denote $s-$value and $p-$value. The results are reported on the $s = -\log_2(p)$ scale because it is intuitive and lends itself to comparison. We return to the advantages of using $s-$value for comparison later. An $s-$value of $k$ can be interpreted as seeing $k$ independent coin flips with the same outcome – all heads or all tails – if the null is that of a *fair* coin [Greenland, 2019]. This conveys how incompatible the data is with the null.

In Figure 1, the case with (1) random sampling exemplifies the type of false alarms we want to avoid. Two-sample classification, standing in for tests of equal distributions, is incompatible with the null of no adverse shift (a $s-$value of around 8). But this shift does not carry over to the other tests. Residual diagnostics, density-based and confidence-based OOD detection are all fairly compatible with the view that the test set is not worse. Had we been entirely reliant on two-sample classification, we may not have realized that this shift is essentially benign. Tests of equal distributions alone give a narrow perspective on dataset shift.

Returning to Figure 1, density-based OOD detection does not flag the (2) in-distribution test set as expected. We might be tempted to conclude that the in-distribution observations are *safe*, and yet, the tests based on residual diagnostics and confidence-based OOD detection are fairly incompatible with this view. Some of the densest points are concentrated in a region where the classifier does not discriminate well: the species 'versicolor' and 'virginica' overlap. That is, the densest observations are not necessarily safe. Density-based OOD detection glosses over this: the trouble may well come from inliers that are difficult to predict. We get a more holistic perspective of dataset shift because of these complementary notions of outlyingness.

## 3  STATISTICAL FRAMEWORK

The theoretical framework builds on Zhang [2002]. Take an i.i.d. training set $X^{tr} = \{x_i^{tr}\}_{i=1}^{n^{tr}}$ and a test set $X^{te} = \{x_i^{te}\}_{i=1}^{n^{te}}$. Each dataset $X^o$ with origin $o \in \{tr(ain), te(st)\}$ lies in $d$-dimensional domain $\mathcal{X} \subseteq \mathbf{R}^d$ with sample sizes $n^o$, cumulative distribution function (CDF) $F_X^o$ and probability density function (PDF) $f_X^o$. Let $\phi : \mathcal{X} \to \mathcal{S} \subseteq \mathbf{R}$ be a score function and define the threshold score $s \in \mathcal{S}$. The proportion above the threshold $s$ in dataset $o$ is in effect the contamination rate $C^o(s) = \Pr(\phi(x_i^o) \geq s)$. The contamination rate is the complementary CDF of the scores, $F^o(s) = 1 - C^o(s)$. As before, $F^o$ and $f^o$ now denote the CDF and PDF of the scores.

Consider the null hypothesis $\mathcal{H}_0 : F_X^{te} = F_X^{tr}$ of equal distribution against the alternative $\mathcal{H}_1 : F_X^{te} \neq F_X^{tr}$. In tandem, consider the null of equal contamination $\mathcal{H}_0^s : C^{te}(s) = C^{tr}(s)$ against the alternative $\mathcal{H}_1^s : C^{te}(s) \neq C^{tr}(s)$ at a given threshold score $s$. To evaluate goodness-of-fit, Zhang [2002] shows that testing $\mathcal{H}_0$ is equivalent to testing $\mathcal{H}_0^s$ for

$\forall s \in \mathcal{S}$. If $z(s)$ is the relevant test statistic for equal contamination, a global test statistic $Z$ for goodness-of-fit can be constructed from $z(s)$, its local counterpart. One such $Z$ is

$$Z = \int z(s) \cdot w(s) \cdot d(s) \qquad (1)$$

where $w(s)$ are threshold-dependent weights. For concision, we sometimes suppress the dependence on the threshold score $s$ and denote weights, contamination statistics and rates as $w$, $z$ and $C^o$ respectively. D-SOS differs from the $Z$ statistic in Equation (1) in three ways: the score function $\phi$, the weights $w$ and the contamination statistic $z$. We address each in turn.

D-SOS scores instances from least to most *abnormal* according to a specified notion of outlyingness. To be concrete, for density estimation, the negative log density is a natural score for outlyingness [Kandanaarachchi and Hyndman, 2021]. This property of the score function $\phi$ can be expressed as

$$\phi(x_i) \leq \phi(x_j) \Rightarrow \Pr(f_X^{tr}(x_i) \geq f_X^{tr}(x_j)) \geq 1 - \epsilon \quad (2)$$

for $x_i, x_j \in \mathcal{X}$ and $\epsilon$, a (sufficiently small) approximation error. Accordingly, instances in high-density regions of the training set $X^{tr}$ (nominal points or inliers) score low; those in low-density regions (outliers) score high. Here, the score function $\phi$ can be thought of as a density-preserving projection. More generally, higher scores, e.g. wider prediction intervals and larger residuals, indicate worse outcomes; the higher the score, the more *unusual* the observation. The structure in $\phi$ is the catalyst for adjusting the weights $w$ and the statistic $z$.

D-SOS updates the weights $w$ to be congruent with the score function $\phi$. When projecting to the outlier subspace, high scores imply unusual points, whereas both tails are viewed as extremes in the Zhang framework. Univariate tests such as the Anderson-Darling and the Cramér-von Mises tests fit the framework in Equation (1): they make different choices for $\phi$, $z$ and $w$. These classical tests place more weight at the tails to reflect the severity of tail exceedances relative to deviations in the center of the distribution. D-SOS corrects for outliers being projected to the upper (right) tail of scores via the score function. The D-SOS weights $w$ are specified as

$$w(s) = \left(F^{tr}(s)\right)^2 \qquad (3)$$

The weights in Equation (3) shift most of the mass from low to high-threshold regions. As a result, D-SOS is highly tolerant of negligible shifts associated with low scores, and conversely, it is attentive to the shifts associated with high scores. Low (high) thresholds map to high (low) contamination rates and low (high) values of the CDF, $F_X^{tr}$. At the lowest threshold, every point in the training set is an outlier; at the highest threshold, none is. Within this spectrum, high thresholds better reflect the assumption that the training set does not contain too many outliers[1]. Thus, we put more weights on high, rather than low, thresholds. Zhang [2002] posits other functional forms, but the quadratic relationship between weights and contaminations as in Equation (3) gives rise to one of the most powerful variants of the test and so, we follow suit.

D-SOS constructs a test statistic based on the score ranks, not on the levels. The weighted area under the receiver operating characteristic curve (WAUC) is a robust statistic that is invariant to changes in levels so long as the underlying ranking is unaffected. The WAUC, denoted $T$, can be written as

$$T = \int F^{tr}(s) \cdot f^{te}(s) \cdot w(s) \cdot d(s) \qquad (4)$$

See Equation (1) in Hand [2009] for details. $T$ in Equation (4) is formally the D-SOS test statistic for adverse shift. The (W)AUC is also threshold-invariant, meaning that it averages (discriminative) performance over thresholds of varying importance. The upside is we do not have to commit to a single threshold. The downside is, if not careful, the (W)AUC summarizes performances over irrelevant regions e.g. low-threshold regions. As a generalization of the AUC, the WAUC puts non-uniform weights on thresholds [Li and Fine, 2010]. D-SOS seizes on this to give more weight to the outlying regions of the data[2]. These data-driven (adaptive) weights conform to the training (reference) distribution, relieving us of the burden of introducing additional hyperparameters.

The null of no adverse shift $\mathcal{H}_0^{ds}$ is that most instances in the test set are not worse than those in the training set. The alternative $\mathcal{H}_1^{ds}$ is that the test set contains more outliers than expected, if the training set is the reference distribution. D-SOS specifies its null as $\mathcal{H}_0^{ds} : T_0 \leq T \mid \mathcal{H}_0$ against the alternative $\mathcal{H}_1^{ds} : T_0 > T \mid \mathcal{H}_0$. $T_0$ is the observed WAUC and $T \mid \mathcal{H}_0$, the WAUC under the null of exchangeable samples. When the test set is cleaner than the training set, tests of goodness-of-fit and of equal distributions both reject the null of no difference, whereas D-SOS by design does not reject its null of no adverse shift. These shifts do not trigger the alarm as they otherwise would had D-SOS been a two-tailed test.

---

[1] We do not assume that the training set only consists of inliers. It may be contaminated, but hopefully not too badly compromised i.e. it is clean enough to be the reference distribution.

[2] Weng and Poon [2008] also advocates for the WAUC as a performance metric for imbalanced datasets because it can capture disparate misclassification costs.

We make a few remarks to further distinguish `D-SOS` from other two-sample statistical tests. Firstly `D-SOS` does not need to specify the equivalence margin, the minimum meaningful difference needed to sound the alarm [Wellek, 2010]. This margin is often a prerequisite of noninferiority tests and similar approaches[3]. Choosing this margin or minimum effect size is often a non-trivial task, requiring substantive domain knowledge [Betensky, 2019]. `D-SOS` sidesteps margin elicitation from experts and relies solely on the training set being an adequate reference distribution. Secondly, tests of equal distributions ignore that the training set precedes the test set, and not vice versa. `D-SOS`, like tests of goodness-of-fit, improve on the former because they use the training set as the reference distribution. Lastly, `D-SOS` deviates from tests of goodness-of-fit. It emphasizes robustness to fight false alarms, it narrows its scope to detecting adverse shifts, not any distribution shift, and it accomodates *any* outlier score[4].

## 4   RELATED WORK

**Outlier scores**. Density ratios and class probabilities often serve as scores in comparing distributions [Menon and Ong, 2016] These scores, however, do not directly account for the predictive performance. Prediction intervals and residuals, on the other hand, do. Intuitively, they both reflect poor predictive performance in some regions of the feature space. Confidence-based OOD detection leans on this insight and tracks *uncertain* predictions e.g. Snoek et al. [2019], Berger et al. [2021]. This is in contrast to density-based OOD detection e.g. Morningstar et al. [2021]. The classical approach, which also accounts for the predictive performance, is based on residual diagnostics and underpin misspecification tests. Janková et al. [2020] is a recent example of this approach with a machine learning twist. Other methods such as trust scores can also flag unreliable predictions [Jiang et al., 2018]. Because all these scores represent distinct notions of outlyingness, the contrasts are often insightful. In this respect, `D-SOS` is in some sense a unifying framework. Bring your own outlier scores and `D-SOS` would morph into the equivalent two-sample test for no adverse shift.

**Dimension reduction**. In practice, the reconstruction errors from dimension reduction can separate inliers from outliers. Recently, Rabanser et al. [2019] uses dimension reduction, as a preprocessing step, to detect distributional shifts. Some methods gainfully rely on the supervised predictions for their low-rank representation [Cieslak and Chawla, 2009, Lipton et al., 2018]. This last approach, to add a cautionary

note, entangles the supervised model, subject to its own sources of errors such as misspecification and overfitting, with dataset shift [Wen et al., 2014]. But it also points to the effectiveness of projecting to a lower and more informative subspace to circumvent the curse of dimensionality. Indeed, the classifier two-sample test uses *univariate* scores to detect changes in *multivariate* distributions [Cai et al., 2020]. Inspired by this approach, `D-SOS` uses outlier scores as a device for dimension reduction.

**Statistical tests**. The area under the receiver operating characteristic curve (AUC) has a long tradition of being used as a robust test statistic in two-sample comparison. Demidenko [2016] proposes an AUC-like test statistic as an alternative to the classical tests at scale because the latter "do not make sense with big data: everything becomes statistically significant" while the former attenuates the strong bias toward large sample size. `D-SOS` is firmly rooted in the tradition of two-sample comparison. Inference is at the sample level. At the instance level, we may want to check if a given test point is an outlier to the training set. Bates et al. [2021] for example tests for individual outliers via conformal $p-$values. Turning pointwise outlier score into an interpretable (standardized) measure such as a probability is a common and useful desideratum [Kriegel et al., 2009, 2011]. Podkopaev and Ramdas [2022], on the other hand, casts the problem of detecting harmful shifts in a sequential setting and controls for the false alarm rate due to continuous monitoring. In sequential settings, statistical process control e.g. [Qiu, 2020] is often used for a similar purpose.

## 5   IMPLEMENTATION

In this section, we turn our attention to deriving valid $p-$values. Without loss of generality, let $\mathcal{D} = [X^{tr}, X^{te}]$ be the pooled training and test set. The score function $\phi = \Phi(\mathcal{D}, \lambda)$ is estimated from data $\mathcal{D}$ and hyperparameters $\lambda \in \Lambda$. This calibration procedure $\Phi : \mathcal{X} \times \Lambda \to \phi$ returns the requisite score function $\phi : \mathcal{X} \to \mathcal{S} \subseteq \mathbb{R}$. The asymptotic null distribution of the WAUC, however, is invalid when the same data is used both for calibration and scoring. We circumvent this issue with permutations, sample splitting, and/or out-of-bag predictions.

Permutations use the empirical, rather than the asymptotic, null distribution. The maximum number of permutations is set to $R = 1000$. We refer to this variant as `DSOS-PT`. Unless stated otherwise, this is the default used in Section 6. For speed, `DSOS-PT` is implemented as a sequential Monte Carlo test, which terminates early when the resampling risk is unlikely to affect the final result materially [Gandy, 2009]. Even so, permutations can be computationally prohibitive.

A faster alternative, based on sample splitting, relies on the asymptotic null distribution. It incurs a cost in calibration accuracy in the first step because it holds out half the data

---

[3]For example, Podkopaev and Ramdas [2022] sets in advance what drop in accuracy, say 5%, constitutes a non-negligible adverse shift.

[4]The score in Zhang [2002] is inspired from the likelihood ratio.

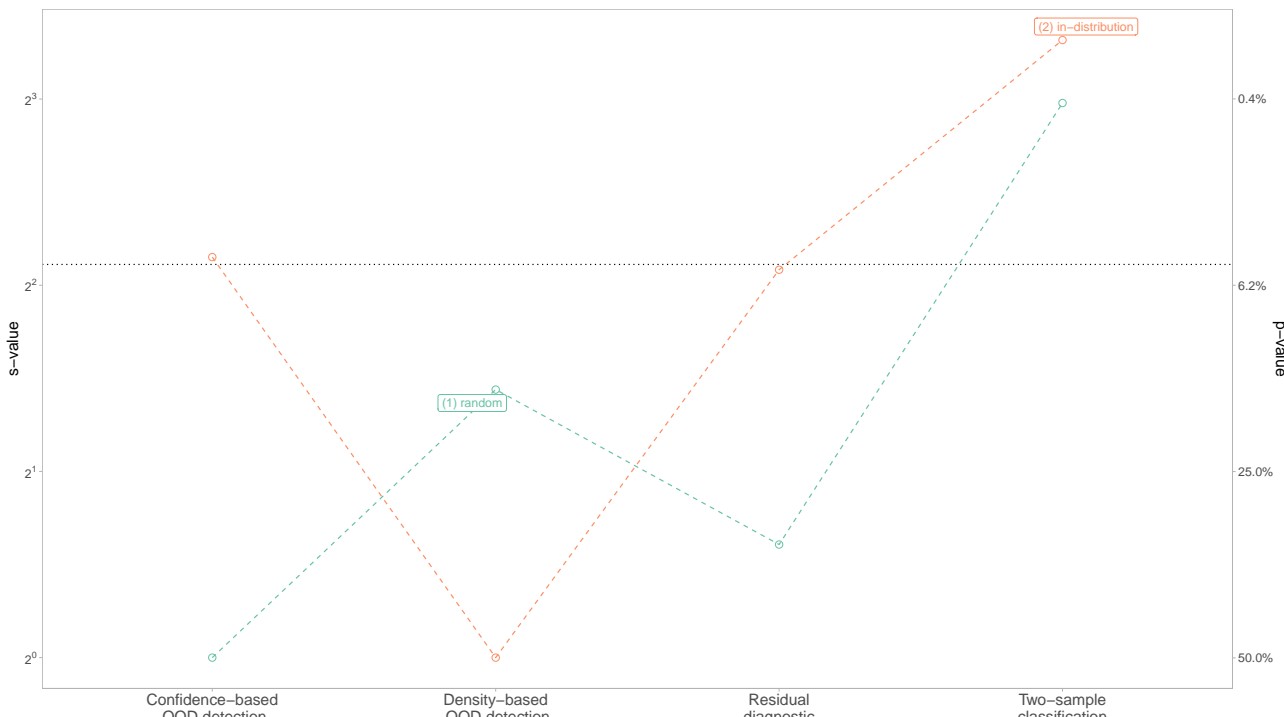

Figure 1: Tests of no adverse shift for `iris`. The tests cover 4 notions of outlyingness for two train-test splits. The x-axis indicates the test type and the colour, the sampling (splitting) strategy. The dotted black line is the common, if not commonly abused, $p-$value $= 0.05$ threshold. We clip $s-$values to a low and high of 1 and 10 respectively and display a secondary y-axis with the $p-$value as a cognitive bridge.

for scoring in the second step. This tradeoff is common in two-sample tests, which requires calibration. We randomly split each dataset in half: $X^{tr} = X_1^{tr} \cup X_2^{tr}$ and $X^{te} = X_1^{te} \cup X_2^{te}$. The first halves are used for calibration, and the second for scoring. We describe this split-sample procedure in Algorithm 1, and refer to it as `DSOS-SS` in Section 6. Given the weights in Equation 3, the WAUC under the null $T \mid \mathcal{H}_0$ is asymptotically normally distributed as

$$T \mid \mathcal{H}_0 \sim \mathcal{N}(\frac{1}{12}, \sigma(n^{tr}, n^{te})) \qquad (5)$$

with mean $= \frac{1}{12}$ and standard deviation $\sigma(n^{tr}, n^{te})$, which depends on the sample sizes. Equation 5 follows trivially from results in Li and Fine [2010] with a little bit of tedious algebra – see Equation (9) therein for the variance.

A third option, an obvious extension to sample splitting, is to use cross-validation instead. In $k-$fold cross-validation, the number of folds $k$ mediates between calibration accuracy and inferential robustness. At the expense of refitting $k$ times, this approach uses most of the data for calibration and can leverage the asymptotic null distribution, provided that the scores are out-of-sample predictions. This strategy is also called cross-fitting in semiparametric inference. It combines the best of both worlds, namely calibration accuracy and inferential speed. We refer to this variant as `DSOS-CV`

in Section 6. We show in simulations that this approach either matches or exceeds the performance of `DSOS-PT` and `DSOS-SS`.

---

**Algorithm 1:** Split-sample test (DSOS-SS)

**Data:** Calibration set $\mathcal{D}_1 = [X_1^{tr}, X_1^{te}]$, Inference set $\mathcal{D}_2 = [X_2^{tr}, X_2^{te}]$, calibration procedure $\Phi$ and hyperparameters $\lambda$

1 **Label:** Assign labels $y = \mathbb{I}(x \in X_2^{te}), \forall x \in \mathcal{D}_2$;
2 **Calibrate:** Fit the score function $\phi = \Phi(\mathcal{D}_1, \lambda)$;
3 **Score:** Score observations $s = \phi(x), \forall x \in \mathcal{D}_2$;
4 **Test:** Compute the observed WAUC $T_0 = \texttt{WeightedAUC}(s, y)$ using Equation (4);
5 **Asymptotic null:** Under the null, $T$ is asymptotically normally distributed as in Equation (5);

**Output:** $p - value = 1 - \Pr(T \le T_0)$

---

## 6 EXPERIMENTS

We make the following pragmatic choices for ease of use; typically, the selected score functions perform well out-of-the-box with little to no costly hyperparameter tuning. For density-based OOD detection, we use isolation forest. To

Table 1: Number of wins for test comparisons based on simulations. [1]Number of ties, not shown, when s-value difference is within ROPE.

| Contender | | Outcome[1] | |
| --- | --- | --- | --- |
| (1) | (2) | Win (1) | Win (2) |
| ctst | energy | 12 | 43 |
| ctst | DSOS-SS | 0 | 34 |
| ctst | DSOS-PT | 0 | 60 |
| ctst | DSOS-CV | 0 | 64 |
| DSOS-CV | energy | 37 | 26 |
| DSOS-SS | DSOS-CV | 0 | 36 |
| DSOS-SS | DSOS-PT | 0 | 32 |
| DSOS-CV | DSOS-PT | 0 | 0 |

investigate other notions of outlyingness, we use random forests. As in Hediger et al. [2022], random forests allow us to use the out-of-sample variant of `D-SOS` (`DSOS-CV`) for free, so to speak. Out-of-bag predictions are viable surrogates for out-of-sample scores. This out-of-bag variant is, when feasible, often convenient as it improves on sample splitting, which sacrifices calibration accuracy for inferential robustness. Details about the experimental setup are in the supplementary material.

## 6.1 SIMULATED SHIFTS

We compare `D-SOS` to two modern tests of equal distribution in unsupervised settings. These benchmarks illustrate the simplicity and performance of our approach where we have an abundance of choice. The field has been churning out new and powerful two-sample tests of equal distribution. These tests are a natural baseline for `D-SOS` because they are commonly used to detect distribution shift. We look at both the `energy` and the classifier test. We show that overall, `D-SOS` performs favorably. We lose little by using a test for no adverse shift instead of a test of equal distribution. `D-SOS` seldom loses and often matches or exceeds the performance of these tests.

The first test is the `energy` test, a type of kernel-based (distance-based) test [Székely et al., 2004]. The second, arguably the main inspiration for `D-SOS`, is a classifier two-sample test, `ctst` [Kim et al., 2021]. `ctst` tests whether a classifier can reliably identify training from test instances. If so, it is taken as evidence against the null. Both `D-SOS` and `ctst` leverage the same underlying classifier and classification scores. As this score increases, so does the likelihood that an observation belongs to the test set, as opposed to the training set. Following Clémençon et al. [2009], this `ctst` variant uses sample splitting, as does `DSOS-SS`, but the AUC rather than the WAUC as a test statistic. The weighting scheme is what differentiates the two.

We simulate distribution shifts from a two-component multivariate Gaussian mixture model and specify distinct shift type (setting): no shift (the base case), label shift, corrupted sample, mean shift, noise shift, and dependency shift. This is described in detail in the supplementary material. When applicable, the shift intensity toggles between low, medium, and high for each shift type. We also vary sample size and dimension. For each combination of shift type, shift intensity, sample size, and dimension, we repeat the experiment 500 times.

To compare the `D-SOS`, `ctst` and `energy` tests, we employ the Bayesian signed rank test [Benavoli et al., 2014]. To do so, we specify a region of practical equivalence (ROPE) on the $s$-value scale and a suitable prior. We deem two statistical tests practically equivalent if the absolute difference in $s$-value is $\leq 1$. This difference corresponds to one more coin flip turning up as heads (or tails), keeping the streak alive. Anything greater arouses suspicion that one test is more powerful, and that the difference is not negligible. Specifying a ROPE on the $s$-value scale is less cumbersome than the $p$-value scale. $p$-values, let alone $p$-value differences, are notoriously difficult to interpret [Wasserstein et al., 2016]. The Bayesian signed rank test yields the posterior probability that one method is better, practically equivalent or worse than the other. The prior for these experiments adds one *pseudo* experiment, distinct from the 500 *real* simulations, where the difference is 0, i.e. no practical difference exists out of the gate.

Table 1 summarizes the findings across all settings: dimension, sample size, shift type and shift intensity. The full tables for comparison are provided as supplementary material. For simplicity, we say that one method wins if the posterior probability that it is better is $\geq 0.5$; similarly, they draw if the posterior probability that they are practically equivalent is $\geq 0.5$. This way of comparing methods resembles the Pitman closeness criterion. We make several observations:

1. `DSOS-PT` and `DSOS-CV` dominate `DSOS-SS`. `DSOS-SS` pays a hefty price in inferential accuracy due to sample splitting. Sample splitting, as evidenced in `DSOS-SS` and `ctst`, clearly trails behind the competition. Even so, `DSOS-SS` outperforms this `ctst` variant: the weights in `DSOS-SS` clearly pay dividends. The WAUC improves upon the plain AUC and offers a compelling alternative to the Mann-Whitney-Wilcoxon (AUC-based) tests when robustness is a chief concern.

2. `DSOS-CV` (or `DSOS-PT`) lag behind `energy` in settings with (1) label shift and (2) corrupted samples. There are 26 such cases. This is because `DSOS` is robust to outliers and more forgiving than nonrobust tests when the bulk of the distributions is largely unaffected. In all other configurations, `DSOS-CV` either wins or draws. In particular, there are 37 cases where

`DSOS-CV` beats `energy`. This is because, like tests of goodness-of-fit, `D-SOS` benefits from conditioning on the the reference distribution while being robust to outliers.

3. When there is indeed a shift, power ($s$-value) increases when sample size and shift intensity increases and decreases when dimension increases, all else equal. This trend is consistent and is as expected. However, the relative efficiency of these tests in small samples and/or at low shift intensity can at times be striking. The 26 cases where `D-SOS` loses to the `energy` test can be characterized as such. In large samples and/or high shift intensity, asymptotics kick in and they all converge.

4. `DSOS-PT` and `DSOS-CV` are practically equivalent across all settings. Notwithstanding the gains in inferential speed that one may hold over the other, the inferential accuracy (power) is – should be – the same. This is a good sanity check that the implementation is correct. Similarly, for the simulations with no distribution shift, all tests are practically equivalent.

### 6.2 PARTITION-INDUCED SHIFTS

In supervised settings, since the core task *is* prediction, the `D-SOS` tests based on residual diagnostics or predictive uncertainty are more informative than, and can be at odds with, classifier (density-based) tests. `D-SOS` creates rigorous statistical tests from these scores so that they can be held to the same standard of evidence as null hypothesis statistical test. This experiment scales beyond the motivational – admittedly, somewhat contrived – example in Section 2 and shows that those findings extend to and are stable across datasets.

To investigate how different notions of outlyingness are correlated, we analyze via cross-validation 62 classification tasks from the OpenML-CC18 benchmark [Casalicchio et al., 2017][5]. As in Section 2, we look at tests of no adverse shift based on two-sample classification, residual diagnostics, and predictive uncertainty (no density-based OOD detection in this round). Cross-validation mimics random sampling variation, and is helpful in assessing stability [Moreno-Torres et al., 2012, Lim and Yu, 2016]; it generates partition-induced shifts, rather than worst-case (adversarial) perturbations [Subbaswamy et al., 2021].

For each dataset, we estimate a fixed (mean) effect reported on the $s-$value scale, which measures how sensitive a dataset is to these partition-induced shifts. The higher the value, the more susceptible a dataset is to adverse shifts caused by sampling variation. These fixed effects can be interpreted in the usual way as the strength of evidence

---

[5]We exclude datasets with more than 500 features because of long runtimes.

against the null of no adverse shift. We refer to the supplementary material for further details about this experiment. At a glance, Figure 2, created with the `superheat` package [Barter and Yu, 2018], tells the story quite succinctly.

Across these 62 datasets, we see that tests of no adverse shift based on two-sample classification, residual diagnostics, and predictive uncertainty (confidence-based OOD detection) are indeed highly correlated. Residual diagnostics and predictive uncertainty, both of which mirror the performance of the underlying supervised algorithm, are the most correlated (Pearson correlation coefficient is 0.82). This suggests that they can be used interchangeably, and by virtue of focusing on predictive performance, are suitable metrics for model monitoring. Two-sample classification is not as strongly associated with residual diagnostics and predictive uncertainty (Pearson correlation coefficient is 0.5 and 0.47 respectively). This arcs back to the point made in Section 2; namely, tests of equal distribution alone are ill-equiped, by definition, to detect whether a shift is benign or harmful for predictive tasks.

Notice the datasets clustered together at the very top of Figure 2 with low $s-$values for two-sample classification but high $s-$values for residual diagnostics and predictive uncertainty. For these datasets, the distribution seems fine but prediction suffers. To a somewhat lesser extent, also notice the reverse effect at the bottom of Figure 2. These are the false alarms we previously highlighted where we have relatively higher $s-$values for two-sample classification than for the other two notions of outlyingness. For these datasets, prediction seems fine but the distribution is not. The link between distribution shift and predictive performance can be more tenuous than what we typically assume. When faced with dataset shift, by rote we often reach for tests of equal distribution. It may be more productive to deliberate over the appropriate notions of outlyingness to detect adverse shift.

## 7 CONCLUSION

`D-SOS` is a wrapper to turn outlier scores into a statistical test. This framework, derived from tests of goodness-of-fit, detects non-negligible adverse shifts. It can confront the data with more pertinent hypotheses than tests of equal distribution. This works well when mapping to the relevant outlier subspace discriminates between safe and unsafe regions of the data distribution. Our method accommodates different notions of outlyingness. It lets users define, via a score function, what is meant by *worse* off. Besides the outlier scores explored in this paper, we stress that other sensible choices for the score functions $\phi$ and the weights $w$ abound. These can be adjusted to account for domain knowledge. Looking ahead, future research could investigate other weighting schemes. The functional form of the postulated weights could be worth tuning. Moreover, *composite* score

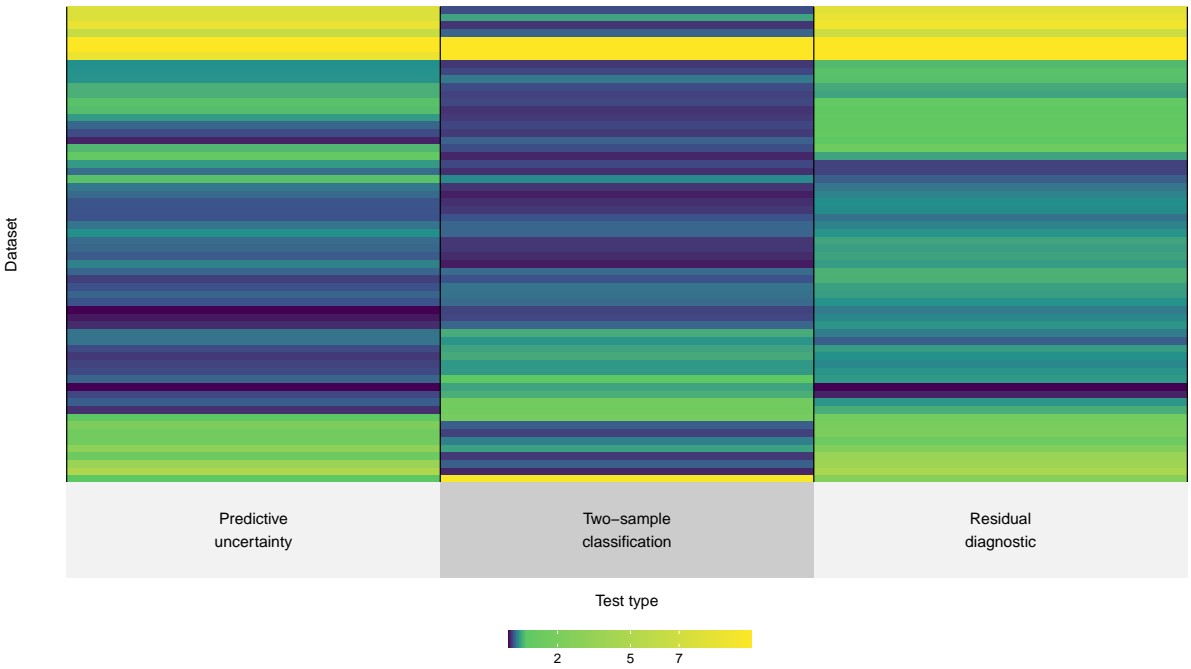

Figure 2: Heatmap of fixed effects for 62 datasets from the OpenML-CC18 benchmark, ordered via hierarchical clustering. Fixed effects are exponentiated and reported on the $s$-value scale for 3 types of `D-SOS` test. The brighter the color, the higher the $s$-value, the more prone the dataset to adverse shift caused by sampling variation.

function, which would combine several notions of outlyingness together, would enrich the types of hypotheses we can test. We imagine situations where we want to combine the strengths of confidence-based and density-based OOD detection into a single aggregate score for example.

### Acknowledgements

The author would like to thank the reviewers for suggesting numerous improvements. Sukanya Srichandra was an excellent sounding board. This work was supported by the Royal Bank of Canada (RBC). The views expressed here are those of the author, not of RBC.

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
