# OpenReview forum: "Test for non-negligible adverse shifts"
_auai.org/UAI/2022/Conference — UAI 2022 Poster_

### Official Review · Reviewer_RGaz · 2022-03-24

**Q2(1) Originality/Novelty:** 3
**Q2(2) Significance/Impact:** 2
**Q2(3) Correctness/Technical Quality:** 3
**Q2(6) Clarity Of Writing:** 3
**Q6 Overall Score:** 6
**Q8 Confidence In Your Score:** 3

**Q1 Summary And Contributions:**

The paper proposes a new approach for testing data distribution shift that incorporates outlier scores. The statistical test is based on the weighted AUC (WAUC) where weights are determined using an existing outlier detection method (so points that score as outliers contribute less). After computing the WAUC using the outlier scores, a p-value is obtained using an approximation of the asymptotic null distribution. Experiments show significant improvements on the OpenML-CC18 datasets.

**Q2 Assessment Of The Paper:**

More detailed information regarding each of these aspects is given below:

**Q2(4) Quality Of Experiments (Optional):**

3: Good: The experimental evaluation is adequate, and the results convincingly support the main claims.

**Q2(5) Reproducibility:**

4: Excellent: Key resources (e.g., proofs, code, data) are available and key details (e.g., proof sketches, experimental setup) are comprehensively described for competent researchers to confidently and easily reproduce the main results.

**Q3 Main Strengths:**

- The approach is based on a simple, straightforward idea that is well motivated

- The empirical results indicate significant improvements over existing methods

- The approach is flexible, different outlier methods may be plugged in

**Q4 Main Weakness:**

- It's not completely clear how similar the approach is to related methods mentioned, i.e. the work appears to by novel, but it's not completely clear from the text

- The experimental results section is very dense whereas the supplement is just a large output of raw results. It's good that comprehensive experiments have been done, but the paper would be easier to read and the impact would be more clear if instead the paper focused on highlighting the most relevant parts of the experimental results with other experiments moved to the supplement.

- It's not clear that the correct baselines are included. It would benefit from more discussion on this.



**Q5 Detailed Comments To The Authors:**

- See weaknesses section above

Additionally, it is surprising that compared method *never* win. Commentary on this might be helpful.

**Q7 Justification For Your Score:**

While the experiments remain not completely clear, the approach is simple, flexible, appears to work well and addresses a limitation of existing methods.

**Q9 Complying With Reviewing Instructions:**

1: Yes.

---

### Official Review · Reviewer_9m1D · 2022-04-12

**Q2(1) Originality/Novelty:** 3
**Q2(2) Significance/Impact:** 2
**Q2(3) Correctness/Technical Quality:** 3
**Q2(6) Clarity Of Writing:** 3
**Q6 Overall Score:** 6
**Q8 Confidence In Your Score:** 3

**Q1 Summary And Contributions:**

This paper presents a new multi-variate two-sample test/framework for detecting dataset shifts. The novel idea of the proposed framework is use outlier scores as the base of the test, such that non-negligible adverse shifts (which have real impact on model predictions), not simply all unequal distributions, can be detected. Experimental evaluation is conducted with simulated shifts and partition-induced shifts with 62 real-world classification tasks.

**Q2 Assessment Of The Paper:**

More detailed information regarding each of these aspects is given below:

**Q2(4) Quality Of Experiments (Optional):**

3: Good: The experimental evaluation is adequate, and the results convincingly support the main claims.

**Q2(5) Reproducibility:**

2: Fair: Key resources (e.g., proofs, code, data) are unavailable but key details (e.g., proof sketches, experimental setup) are sufficiently well-described for an expert to confidently reproduce the main results.

**Q3 Main Strengths:**

1. Interesting and novel idea of turning outlier scores into a two-sample test for detecting adverse shift.
2. Good work on illustrating the proposed framework and the motivating example.
3. Solid experimental evaluation.

**Q4 Main Weakness:**

1. More detailed justification/discussion on some of the key details is needed.
2. The presentation could be improved to make the writing more explicit overall.

**Q5 Detailed Comments To The Authors:**

More detailed justification/discussion on the key technical details, including
1. Justification the use of WAUC as the test statistic.
2. The impact of the outlier threshold score s.
3. While the proposed framework reduce false positive rate, how it would assure low false negative rate?

Presentation:
Overall the writing is easy to follow, but sometimes it could be made more explicit and/or the key idea/points could be presented upfront or addressed more clearly. For example, the very last paragraph of Introduction (the "takeaway" message) is well written, but it would be helpful if the idea can be pointed out earlier in an explicit way like in this paragraph.

**Q7 Justification For Your Score:**

The idea of the proposed framework sounds interesting and novel. The focus on non-negligible adverse shifts has high practical value. Overall I find the method is well motivated and well designed, and the experiments are solid too. Some technical details, however, should be discussed to strengthen the contribution and let readers know the restrictions of the method. Presentation of the paper can be improved too.

**Q9 Complying With Reviewing Instructions:**

1: Yes.

---

### Official Review · Reviewer_FNK2 · 2022-04-13

**Q2(1) Originality/Novelty:** 1
**Q2(2) Significance/Impact:** 2
**Q2(3) Correctness/Technical Quality:** 3
**Q2(6) Clarity Of Writing:** 4
**Q6 Overall Score:** 6
**Q8 Confidence In Your Score:** 4

**Q1 Summary And Contributions:**

The paper presents a hypothesis test for adverse distribution shifts. The proposed method allows the users to define their own scoring function for different aspects of distribution shifts, and use the weighted AUC as the statistic.

**Q2 Assessment Of The Paper:**

More detailed information regarding each of these aspects is given below:

**Q2(4) Quality Of Experiments (Optional):**

2: Fair: The experimental evaluation is weak: important baselines are missing, or the results do not adequately support the main claims.

**Q2(5) Reproducibility:**

4: Excellent: Key resources (e.g., proofs, code, data) are available and key details (e.g., proof sketches, experimental setup) are comprehensively described for competent researchers to confidently and easily reproduce the main results.

**Q3 Main Strengths:**

The paper presents a new method for testing for distribution shifts, that can focus on different aspects of such shifts defined by the user
The paper is very clearly written and easy to follow
The problem is important

**Q4 Main Weakness:**

The method has limited novelty: The main components of the method are the scoring function which is user-defined an using WAUC instead of AUC, none of which is entirely new.
The motivation for the choice of weights (Eq. 3) is not entirely clear to me.
Since the theoretical results of the paper are not so strong or novel, I would perhaps like to see a much larger experimental evaluation, showing how the method compares to existing approaches in various different scenarios. Instead, the experimental results seem to illustrate that different scores lead to different results, which seems fairly intuitive.

**Q5 Detailed Comments To The Authors:**

What is the motivation behind the weights? Perhaps presenting an example where this would be useful vs plain AUC would be helpful.
For the experimental section, showing some case studies where the method can reject a null hypothesis that would not be rejected by related work methods could help. As it is, I am not entirely sure this paper presents an entirely novel method or if it is a collection of multiple existing methods.

**Q7 Justification For Your Score:**

While the theoretical results are not very strong, the method could potentially be useful.

**Q9 Complying With Reviewing Instructions:**

1: Yes.

---

### Decision · Program_Chairs · 2022-05-15

**Decision:**

Accept (Poster)

**Comment:**

Meta Review: Although the main idea is relatively incremental and combines existing ideas, the problem tackled is particularly relevant to UAI and the manuscript does a solid and transparent job in describing its contribution, and how it adds to the literature.